# A Quality of Experience assessment of haptic and augmented reality feedback modalities in a gait analysis system

**Thiago Braga Rodrigues**[1]*, **Ciarán Ó Catháin**[2], **Noel E. O'Connor**[3], **Niall Murray**[1]

**1** Faculty of Engineering & Informatics, Athlone Institute of Technology, Athlone, Westmeath, Ireland, **2** Faculty of Science & Health, Athlone Institute of Technology, Athlone, Westmeath, Ireland, **3** Insight Centre for Data Analytics, Dublin City University, Dublin, Ireland

* t.brodrigues@research.ait.ie

**Data Availability Statement:** All relevant data are within the paper and its Supporting Information files.

## Abstract

Gait analysis is a technique that is used to understand movement patterns and, in some cases, to inform the development of rehabilitation protocols. Traditional rehabilitation approaches have relied on expert guided feedback in clinical settings. Such efforts require the presence of an expert to inform the re-training (to evaluate any improvement) and the patient to travel to the clinic. Nowadays, potential opportunities exist to employ the use of digitized "feedback" modalities to help a user to "understand" improved gait technique. This is important as clear and concise feedback can enhance the quality of rehabilitation and recovery. A critical requirement emerges to consider the quality of feedback from the user perspective i.e. how they process, understand and react to the feedback. In this context, this paper reports the results of a Quality of Experience (QoE) evaluation of two feedback modalities: Augmented Reality (AR) and Haptic, employed as part of an overall gait analysis system. The aim of the feedback is to reduce varus/valgus misalignments, which can cause serious orthopedics problems. The QoE analysis considers objective (improvement in knee alignment) and subjective (questionnaire responses) user metrics in 26 participants, as part of a within subject design. Participants answered 12 questions on QoE aspects such as utility, usability, interaction and immersion of the feedback modalities via post-test reporting. In addition, objective metrics of participant performance (angles and alignment) were also considered as indicators of the utility of each feedback modality. The findings show statistically significant higher QoE ratings for AR feedback. Also, the number of knee misalignments was reduced after users experienced AR feedback (35% improvement with AR feedback relative to baseline when compared to haptic). Gender analysis showed significant differences in performance for number of misalignments and time to correct valgus misalignment (for males when they experienced AR feedback). The female group self-reported higher utility and QoE ratings for AR when compared to male group.

**Funding:** The work presented in this paper has been supported by the Irish Research Council under grant GOIPG/2017/803 awarded to T.B.R. This publication has also been supported by the Science Foundation Ireland (SFI) under grant number SFI/12/RC/2289_P2 awarded to N.OC. and grant number SFI/13/RC/2106 awarded to N.M. The funders had no role in study design, data collection and analysis, decision to publish, or preparation of the manuscript.

**Competing interests:** The authors have declared that no competing interests exist.

# 1 Introduction

The assessment of human gait facilitates identification of movement deficiencies and abnormalities that are associated with the development of chronic injuries and disease. It provides objective data to support rehabilitation and retraining. Gait can be analysed and assessed using a variety of methods such as: clinical evaluation techniques; the use of high-speed cameras; force plates; and inertial sensors [1]. The hip and knee are weight bearing joints and play a key role in gait stability. The displacement of knee—called varus/valgus—is a misalignment of the tibiofemoral joint. The valgus knee (as per Fig 1a) is a condition whereby the knees turn outwards, whilst in the varus knee (Fig 1c) is a condition that causes the knees to turn inwards inwards [2]. This disorder occurs because the tibia is not aligned correctly with the femur, giving a different shape to the leg line.

Excessive varus/valgus alignment can lead to serious orthopedics problems such as osteoarthritis [3]. Extreme cases of knee misalignment may need to be addressed surgically. If not properly treated, it can result in severe injuries from joint wear to diseases, e.g. knee arthrosis and osteoarthritis. However, in less severe cases, symptoms can be reduced with physiotherapy, corrective exercises, and through gait re-training [4]. There are some rehabilitation procedures to help with varus/valgus knee such as strengthening of hip and knee muscles [5]. Critical to all types of rehabilitation is appropriate feedback.

Feedback is a powerful tool for motor skill learning and helps with the sensory perceptual information as part of performing and learning a skill [6]. The accuracy of exercise performance with feedback in physiotherapy influences the healing process of the patient greatly. Crucial to successful rehabilitation is for the patient to understand the feedback, be it from a clinician or system. [7]. Some of the feedback systems include modalities such as: 2D screens; haptic; audio; expert guidance; and in more recent times Virtual Reality (VR) and Augmented Reality (AR) [8–10].

All of the different feedback approaches have advantages and disadvantages. For example, with 2D screen feedback, the user is limited in terms of the direction they can walk i.e. they are

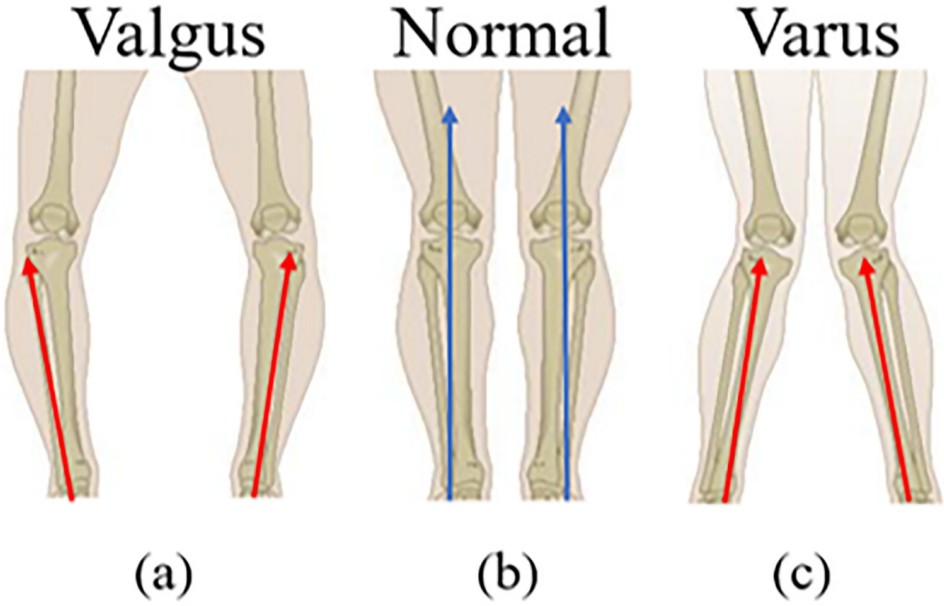

**Fig 1. Tibia alignment: Varus (1a), normal (1b), and varus (1c) knee.** Red arrows represent misalignment in the tibiofemoral joint. The blue arrows represent alignment of the tibiofemoral joint.

always required to walk towards the screen and must have their head up facing the screen. Audio guidance addresses this issue, but users need to clearly understand the guidance. With audio, this has shown to be an issue and a source of confusion [8]. The use of expert guidance has many benefits, but it requires the user to attend an expert clinic; the expert to be available; and is based on subjectivity of the clinician. Considering existing feedback modalities, 2D screens, audio and expert guidance, haptic has been shown to support the greatest user improvement for gait re-training [11, 12]. However, this requires the accurate placement of the haptic feedback display. AR has been very successful in education and considering the limitations of existing approaches, and the potential of AR as a portable, wearable and visual feedback modality is under researched and certainly worth investigating.

AR is an interactive experience in a real-world environment whereby real world objects are augmented with virtual information [13]. The future of AR points to a deeper use of technology augmenting human performance across a range of application domains [14]. Despite the countless possible applications and advances in the industrial sector, the understanding of user perception of AR technology is limited. Hence, there is a real need for user studies to determine the usability and utility of AR in different domains. This can be addressed through the Quality of Experience (QoE) framework. In this work we employed a questionnaire in order allow users to self-report on their perception of AR and Haptic feedback systems (in terms of utility, usability, interaction, and immersion). In terms of system utility (e.g. easiness to adjust to feedback), usability (e.g. feedback easy to understand), interaction (e.g. how users interact with feedback), immersion (e.g. awareness of body whilst moving.). The use of AR via wearable smart glasses in the field of gait rehabilitation is certainly an area under researched to-date. This study investigates if AR has the potential to be a lightweight and portable feedback alternative for rehabilitation protocols considering both objective (performance) and subjective (user QoE) evaluations.

Whist the previous discussion has justified the importance of understanding user perceptual quality of haptic and AR based gait feedback, the task of measuring user perceptual quality of multimedia experiences is complex. QoE is a user centric paradigm that allows us to evaluate the "degree of enjoyment or annoyance of an application, system, or service" of a multimedia experience [15]. It represents "the fulfillment of user's expectation in respect to utility and enjoyment of that application or service" [16]. In order to evaluate any service and system from a QoE perspective, different Influencing Factors (IF) need to be considered. There are three main IF categories in QoE research: Human IF (e.g. gender, background), Context IF (e.g. Physical condition of varus/valgus in the case of this work, task), and System IF (e.g. AR versus Haptic, colour, screens).

In the recent years, with the advent of internet, advanced sensors, and internet of things (IoT), new proposals on evaluating QoE in a continuous manner have been proposed [17, 18], and models of assessing several multimedia systems were built [19]. Here, with the utility of the feedback as a key concern, the proposed work presents a novel QoE "system" level comparison of two feedback modalities (AR vs Haptic) within a gait analysis system. Our QoE comparison includes data analysis from post-test self-reported measures and also objective data comparison in terms of user responses (i.e. changes in gait if any) to each of the feedback modalities. In addition, we include analysis on the human factor (gender) and its effects on QoE and performance.

## 2 Related work

This section contains a critique of related research in terms of multimodal gait feedback systems and QoE assessments approaches for Haptic and AR (not all are specific to gait feedback). Each of these aspects are relevant to the scope of this work.

Haptic feedback has been studied in many works related to human activities [20], motor learning [21], and gait retraining [22]. Numerous works have compared haptic feedback with other modalities and have reported haptic: to be "less intrusive" than virtual reality feedback [23]; to be better in supporting task performance when compared to visual feedback for lower extremities [24] in gait; not to affect ecological validity of interaction compared with other modalities [25]. In addition, Haptic has been reported as easier to understand and follow when compared to auditory and visual stimuli [26]. Haptic feedback was also used to enhance the realism of a walking experience in multimodal environments [27]. Haptic feedback has also been used as an important tool in gait retraining for treatment of knee osteoarthritis [28]. In [11], closely aligned to the focus of this work, a gait re-training system employed haptic feedback to change gait parameters including varus/valgus misalignments. The system and results served as basis for this work by informing the use of haptic feedback to capture and improve gait parameters including knee alignment. They also highlighted issues whereby users were confused when receiving more than one feedback simultaneously (i.e. on different parts of the body). Such issues are again validation for why QoE assessments of such feedback mechanisms are required.

Some authors have applied Augmented Reality in gait analysis. In [29], a low-cost gait analysis system was developed using AR markers and a single video camera. The AR markers were used to track body segments and capture gait variables. Even though the authors achieved calibration and accurate tracking for gait angles, they highlighted the use of markers as a limitation (e.g. this system could not be used for treadmill walking). The use of different AR devices was also reported for guided walking in [30, 31]. These works indicated that novel AR technologies could be used in walking guidance with performance, body stability with positive impact in gaze and locomotor control [32, 33].

Considering these works, the use of AR for gait feedback has not been deeply explored. There are some exploratory works that suggest employing AR for gait retraining. The results reported in [34] results in significant improvement in gait over a 2D monitor. Other research has reported the use of AR in gait posture training [35] reported statistically significant improvement in posture, balance, and velocity. In [36], a gait retraining system was developed to modify footprint parameters. The authors concluded that AR could help to quickly modify user's footprint parameters. Although these works make a valuable contribution, there was no qualitative metric employed that informs if users were satisfied or enjoyed the feedback experience. This is critical because it informs designers about how the users enjoy, engage, and experience such systems.

Several authors have used QoE assessment in multimedia systems as a paradigm to quantify how various factors of the system influence perceived quality levels from the user perspective. In [37], user QoE levels were compared in an immersive Virtual Reality and Augmented Reality applications. A sample size of twenty-one participants was divided randomly into two groups. Both objective and subjective metrics were gathered. The authors considered system, psychological, and user factor to evaluate quality. The QoE evaluation suggested that users felt safer and accustomed with the use of AR when compared to virtual reality. In [38], a QoE evaluation of a motor skills rehabilitation game was developed. The authors have assessed QoE through user engagement, task success, interaction, and socialization. This study reported that high QoE scores can be linked to high performance. These works demonstrate the need of a qualitative study for different applications. Several works have reported that valgus/varus incidence is different across gender groups [39, 40]. Anatomical differences between males and females lead to differences in knee alignment, and are a potential cause of anterior cruciate ligament injuries in females [41, 42]. Females, in general, have wider hips than males, influencing kinematic factors related to injury such as knee valgus/varus. Since the incidence of valgus/

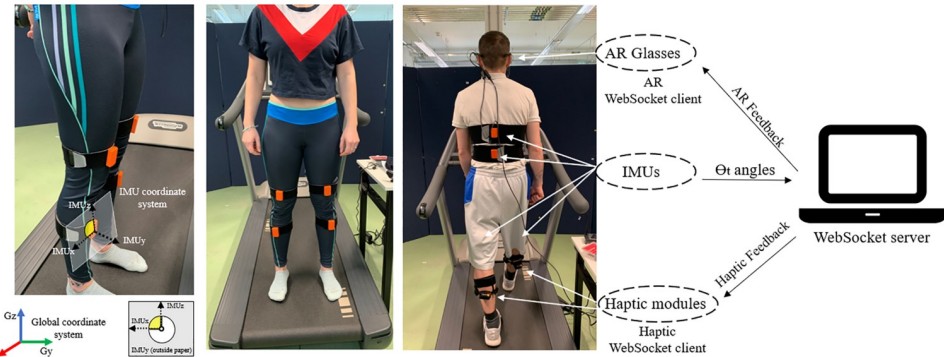

**Fig 2. Gait feedback system modules and system architecture.** The figure shows sensor placement and coordinate systems from different views.

varus misalignments is different between males and females, we considered gender as an important factor to consider of this study.

Considering existing literature, the novelty of the work presented in this article lies in the evaluation and analysis of users' QoE (self-reported measures and objective measures) of Haptic and AR feedbacks in our gait analysis system. The focus is on comparing subjective and objective metrics for correcting knee alignment with these two different feedback modalities (Haptic and AR).

## 3 System and feedback architecture

Our gait system is composed of a capturing module, a presentation module and a data processing module. The capturing module consists of 6 Inertial Measurement Units (IMUs). The Feedback module contains two components: Haptic and Augmented Reality modules. Finally, the data processing system is a quadcore Intel Core i7 laptop, 16GB DDR4 RAM, 3.2Ghz, GTX 1060-6GB was used to integrate all modules and is also the Wi-Fi WebSocket server for all modules as per Fig 2.

### 3.1 Capturing module—IMU

The capturing module contains 6 X-Sens IMU's [43] and placed on the body as per Fig 2. A real-time Wi-Fi synchronization and streaming protocol for multiple IMUs was developed in C#. This streaming protocol is important for it ensures that no data is lost, that feedback is presented without delay, and all modules can work independently. In terms of internal configuration of each IMU, 10 streams of data were captured: 3D acceleration from triaxial accelerometer ($Acc_{xyz}$), 3D angular velocity from triaxial gyroscope ($Gyro_{xyz}$), 3D magnetic field from a triaxial magnetometer ($Mag_{xyz}$), and UNIX timestamp. As discussed later in this section, the $Acc_{xyz}$, $Gyro_{xyz}$, and $Mag_{xyz}$ were fused to provide quaternion representation. The developed protocol fuses, in real-time, accelerometer, gyroscope, and magnetometer data and generates the quaternion orientation. The datasets from the IMU's were synchronized with the computer CPU clock ensuring no packet loss. This module, processes in real time, the quaternion and Euler angles of each sensor and generates angles for knees, hips, tibia, and trunk lean. Data from the sensors was sampled at 40Hz on all three axes and sent through a Wi-Fi interface to the server computer. Further details on the multi-IMU streaming protocol is available in [44, 45] for the interested reader.

To represent the orientation of a rigid body or frame coordinates in 3D space, a quaternion representation was employed. This complex number representation defines any spatial rotation around a fixed point or coordinate system. A quaternion $q = [q0q1q2q3]$ was used to calculate an angle $\theta$ about a fixed Euler axis [46, 47]. To get the angle between two joints with IMU, quaternion matrices were obtained by fusion of the 3 internal modules ($Acc_{xyz}$, $Gyro_{xyz}$, $Mag_{xyz}$) using a Madgwick-based orientation filter [48]. The quaternion generated by the orientation filter represent s the spatial rotation of each IMU and can generate any joint angle (knee angle in this case) for each axis. Having each Euler angle, it is then possible to reference one IMU to another and determine the angle between two sensors. This angle between the two IMU's was used as part of the walking evaluation during experiments. At the start of each test, while the user was stand, sensor calibration was obtained using the IMU quaternion in Euler angles $\theta_x$, $\theta_y$, $\theta_z$ in North-East-Down (NED) Z-Y-X sequence as in Eq (1).

$$
\begin{bmatrix} \theta_x \\ \theta_y \\ \theta_z \end{bmatrix} = \begin{bmatrix} arc\tan \dfrac{2(q_0q_1 + q_2q_3)}{1 - 2(q_1^2 + q_2^2)} \\ arc\sin\left(2(q_0q_2 - q_3q_1)\right) \\ arc\tan \dfrac{2(q_0q_3 + q_1q_2)}{1 - 2(q_2^2 + q_3^2)} \end{bmatrix} \tag{1}
$$

To find the tibia projection angle in the frontal, lateral, and sagittal planes, we need to calculate unit vectors on each quaternion coordinate system. This calculation converts the current quaternion of each IMU to direction cosine matrices. We take then the calibrated $\theta_x$, $\theta_y$, $\theta_z$ and convert them into a unit vector in the ZYX order as in Eq (2). We then applied this to the calibrated Euler angles.

$$
\begin{bmatrix} IMU_x \\ IMU_y \\ IMU_z \end{bmatrix} = \begin{bmatrix} M[1,1] & M[1,2] & M[1,3] \\ M[2,1] & M[2,2] & M[2,3] \\ M[3,1] & M[3,2] & M[3,3] \end{bmatrix} given,
$$

$$M[1,1] = Cos(\theta_y)Cos(\theta_z)$$

$$M[2,1] = Cos(\theta_z)Sin(\theta_x)Sin(\theta_y) + Cos(\theta_x)Sin(\theta_z)$$

$$M[3,1] = -Sin(\theta_x)$$

$$M[1,2] = -Cos(\theta_y)Sin(\theta_z) \tag{2}$$

$$M[2,2] = Cos(\theta_x)Cos(\theta_z) - Sin(\theta_x)Sin(\theta_y)Sin(\theta_z)$$

$$M[3,2] = Cos(\theta_y)Sin(\theta_x)$$

$$M[1,3] = Sin(\theta_z)$$

$$M[2,3] = -Cos(\theta_y)Sin(\theta_x) + Cos(\theta_x)Sin(\theta_y)Sin(\theta_z)$$

$$M[3,3] = Cos(\theta_x)Cos(y)$$

To get any IMU joint angle (tibia angle in our case), we convert each IMU quaternion into Direction Cosine Matrices ($DCM_{xyz}$) (Eq (3)) and multiply the direction vector $IMU_{xyz}$ as in Eq (4). Finally, we apply trigonometry of right-angle triangle of $IMU_{xyz}$ of the directional

vector $v_{xyz}$ on the desired plane (Eq (5)). The angle $\theta$ between two IMU will be as in Eq (6).

$$DCM_{xyz} = \begin{bmatrix} (q_0^2 + q_1^2 - q_2^2 - q_3^2) & 2(q_1q_2 + q_0\,q_3) & 2(q_1q_3 - q_0\,q_2) \\ 2(q_1q_2 - q_0\,q_3) & (q_0^2 - q_1^2 + q_2^2 - q_3^2) & 2(q_2q_3 + q_0\,q_1) \\ 2(q_1q_3 + q_0\,q_2) & 2(q_2q_3 - q_0\,q_1) & (q_0^2 - q_1^2 - q_2^2 + q_3^2) \end{bmatrix} \tag{3}$$

$$v_{xyz} = [DCM]\begin{bmatrix} IMU_x \\ IMU_y \\ IMU_z \end{bmatrix} \tag{4}$$

$$\theta_{IMU} = atan\left(\frac{v_x}{v_z}\right) \tag{5}$$

$$\theta = \theta_{IMU1} - \theta_{IMU2} \tag{6}$$

## 3.2 Feedback modules

In this section, Haptic and Augmented Reality feedback modules are presented.

**3.2.1 Haptic module.** A bespoke wearable haptic module was designed for gait feedback purposes as illustrated in Fig 3. No off-the-shelf haptic modules satisfied our requirements of being lightweight, wearable, and provide a haptic sensation. The Haptic module was developed to provide the correct feedback to the user according to his/her movements [49]. The two haptic modules had an ESP8266 Wi-Fi micro-controller board with a WebSocket client. Each module was composed of a leg mounted strap; two vibration units (Fig 3a); and communication and micro-controller with battery unit (Fig 3b).

The leg mounted bracelet is attached to the users' skin as per Fig 2. The vibration units are enclosed within the plastic casing. The design of the circuit contains MOSFET transistors

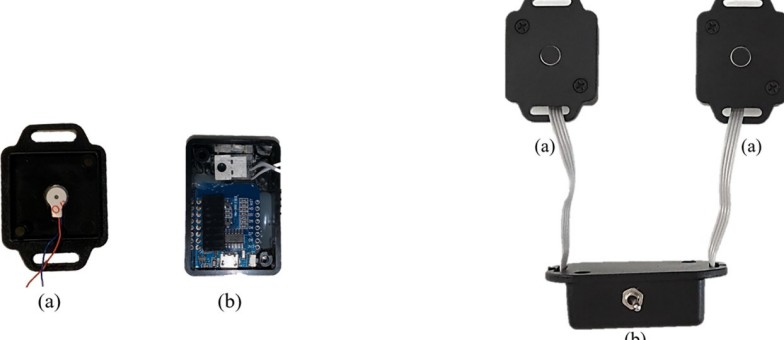

(a)   (b)

(a)   (a)

(b)

**Fig 3. Haptic feedback module.** It contains haptic motors (a) and the Wi-Fi microcontroller responsible for the web-socket client (b). All the units are sheltered within ABS plastic cases (30x30x10mm) for the haptic module and (40x30x10mm) for the Wi-Fi micro-controller.

operating as switches. There was also a pulse width modulation control to allow precise change of the intensity of the vibration unit if required. When the signal is received by the communication unit, the vibrating unit provides a high level TTL output signal to the transistor's gate. This signal leads the transistor to operate in the "saturation region" and permitting the current to reach the motor. A freewheel diode was installed across each motor of the vibration units to remove voltage spikes due inductive nature of the load when switched off [50]. This prevents malfunction of the hardware, protecting the I/O ports of the microcontroller inside the communication unit from electromotive force (EMF).

**3.2.2 Augmented reality module.** Our AR module consisted of an Epson Moverio Bt-300 Smart Glasses [51] connected with a WebSocket protocol. A WebSocket client in the AR module was employed as it allowed the web server to establish a connection with the feedback application and communicate directly with it without any delay (typically web communication consists of a series of requests and responses between the client and the web server, where, for real-time applications, this technique is not well suited [52]). With the use of WebSockets, we established a connection only once, and the communication between the server and the feedback application could follow without problems related to delay and synchronization.

**3.2.3 Activation of feedback modules.** The feedback state diagram is shown in Fig 4. The user input is compared with the kinematic model which controls the feedback mechanism according to the activation threshold. The kinematic model was defined as per Fig 4, with activation thresholds for each feedback defined at $+7^o$ for valgus, and $-7^o$ for varus i.e. if valgus/ varus angle extended beyond the defined threshold, feedback was provided to the user. These values represent normal angle limits of knee alignment [53]. The model constantly evaluates the current tibia angle in order to compare with threshold values. Each person has their own walking style and for this reason it is difficult for a participant to have perfect alignment throughout every single part of the gait cycle while walking naturally. Because of this, every small change between baseline (no feedback) and test (both feedback) was observed during testing.

The feedback in the Haptic module was presented as vibrations on each leg whenever the participant's tibial angle was above or below the activation thresholds for valgus and varus. The correct alignment of each leg resulted in "no vibration" (i.e. no feedback provided) on the Haptic bracelet. During the training phase (see section IV), participants were told that no feedback from haptic means they are in correct alignment. The objective given to the participant was to receive the least amount of vibration as possible. The feedback in the AR module was

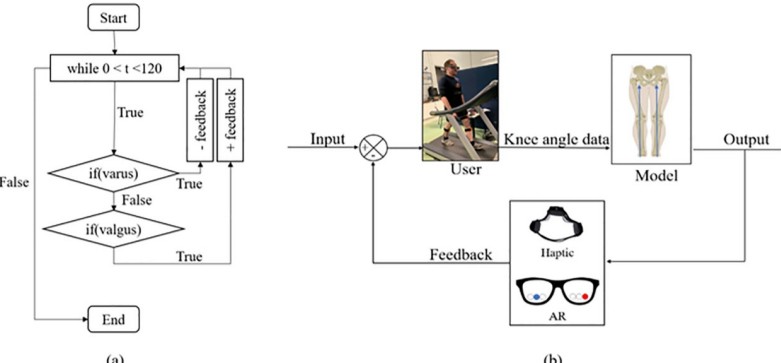

**Fig 4. Flowchart (a) and feedback state diagram (b).** These diagrams represent the feedback control system. User knee angle is used as input, which will be compared constantly with kinematic model. The user then receives haptic or AR stimuli to correct knee alignment.

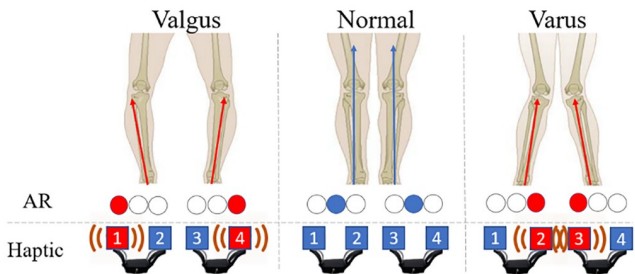

**Fig 5. AR and haptic feedback activation controls.** AR feedback iscontrolled by colored circles: redfor misalignments and blue for alignment. Haptic controls are vibrations oneach leg: 1 and 4 for Valgus, 2 and 3 for Varus.

presented as circle visualizations on the AR glasses (see Figs 4 and 5). The user sees a projection of 6 circles in their field of view (3 of each leg as per Fig 5). Again, whenever the tibial angle was above or below the activation thresholds for valgus and varus. For each leg, three circles control the states of the knee according to valgus and varus angles. The correct alignment of each leg is achieved when the blue circle in the middle is lit. The objective given to the participant is to keep the circles blue during trial.

## 4 Experimental protocol

This research was approved by the Athlone Institute of Technology Research Ethics Committee on the 23rd of January of 2019. Participants consent was obtained in written format and stored in a secure location. Data were anonymized for all trials and participants. After ethics approval, a test with healthy participants was conducted. A convenience sampling approach was employed to recruit twenty-six participants (13 males, 13 females) with an average age of 27.54 (± 6.57) years. Due to previous knee or walk abnormalities, data of two participants was omitted. The gender balance guidelines have been applied as per ITU-P913 standards for objective and subjective quality assessment [54]. A within group experimental design was employed; hence each participant experienced both the haptic and AR feedback modalities. The ordering of how the participants experienced the feedback was randomised. Participants were tested on two different days and the protocol adhered to the approach taken in numerous related works in the literature [17, 37, 55] and included the steps outlined in Fig 6.

During the information phase, each participant was greeted and thanked for their participation. After a brief explanation, written consent was obtained. Participants were brought to the waiting room and were provided with an information sheet that fully described the experiment. The screening phase assessed a participant's visual acuity color perception, and ability to perceive the haptic stimuli [56–58]. The screening process for participants for visual acuity, color perception, and haptic sensation required participants to achieve a threshold score to be eligible for the actual testing. For the Snellen test, a score of 20/20 was required. For the Ishihara test, thirty-eight color plates were used and only 4 errors were allowed during examination. For the haptic screening, participants were required to differentiate 4 vibration patterns and location [58]. Upon completion, baseline metrics of gait angles: left and right hip, left and right knee, left and right tibia (for varus/valgus assessment), and trunk lean were captured over a two-minute period using the devices outlined in Section III. For this experiment, we only analyzed tibia angle to evaluate feedback. Full gait analysis considering all angles will be evaluated as part of a future work study.

For training and testing phases, participants were randomly assigned into two groups (Haptic/AR, and AR/Haptic) depending on which feedback the participant experienced first. Each

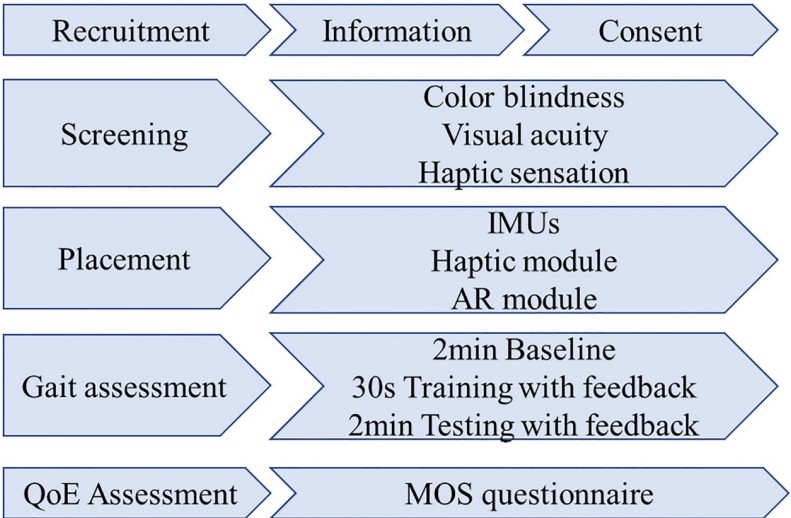

**Fig 6. Testing protocol.** This protocol was consisted during all trials for all participants. The full protocol is available in S1 File.

participant experienced one of the feedback modalities and had a week break before they were presented with the alternative modality feedback. As part of the training, participants were introduced to the AR and the haptic modules as appropriate for the given test day. The devices were fitted to the participant by the principal investigator and an opportunity for adjustment was provided to ensure there was no discomfort. After sensor placement, participants were securely guided to a treadmill where they were asked to select a walking speed with which they felt comfortable (the range selected by users was between 2.5 and 4 miles per hour). Following this, in the test, the speed each participant selected was maintained for training and testing of both feedback modalities. Instructions for each feedback were explained with 3 feedback sheets (available in S1 File) showing the difference of the three different knee states (valgus, normal, varus). Participants were aware that each leg was independent so that even though one leg was on valgus state, the other one could be aligned for example. Participants walked 2 minutes for base-line capture (no feedback), 30 seconds for feedback training, and 2 minutes (with feedback).

## 4.1 QoE questionnaire

As per [59], twelve questions asked were asked of all participants on the experience of both feedback modalities. For the subjective analysis, QoE factors were evaluated in form of questionnaires after the gait assessment phase as per Fig 6. QoE takes into consideration how system, human and contextual factors contributes to a user's perceived quality of a system [19]. The literature suggests that the accepted approach to measuring a user's perceived quality of his or her experience is based on self-reported measures via post-experience questionnaires. The developed questionnaire was used to determine an overall mean opinion score (MOS) based on feedback from users [60].

The twelve questions were developed to evaluate system utility (questions 1-3), usability (questions 4-6), interaction (questions 7-9), and immersion (questions 10-12). For each of those 4 assessment variables, 4 standard questionnaires were used as guidelines: The System Usability Scale (SUS), ITU-T methods for subjective assessment of quality, Igroup presence questionnaire (IPQ), and Computer System Utility Questionnaire (CSUQ) [54, 61–63]. The

rating system used was a seven-point Likert scale to determine whether or not the participant agreed with the statement. The full questionnaire is available in [59] and per Table 3 in the results section. The ordering of the questions was randomized for the different participants to negate any ordering effects.

## 4.2 Data processing and statistics

As outlined in the methodology section, QoE and objective metrics were captured for each trial. Participants were categorized into AR and haptic. Subgroups of males (N = 13) and females (N = 13) were also randomly defined for gender analysis purposes. In order to compare differences across groups, a Shapiro-Wilk normality test [64] was conducted. All variables were with a normal distribution ($p > 0.05$). A dependent samples t-test was performed on the data with 95% confidence level. For the objective analysis, we have reported differences between AR and haptic groups for number of alignments after receiving feedback, and the amount of time participants were not aligned. We have also reported the same analysis considering gender. These comparisons were done by dependent samples t-test at 95% confidence level. The QoE model ($QoEM_F$) for each feedback for a number p of participants was designed to be average of the four-assessment metrics: Utility (UtF), Usability (UsF), Interaction (InF), and Immersion (ImF) as in Eq (7).

$$QoEM_F = \sum_{n=1}^{p} \frac{UtF_n + UsF_n + InF_n + ImF_n}{4} \qquad (7)$$

# 5 Results

In this section we present analysis and discussion of the data captured during the experiment: objective measures of performance (i.e. number of misalignments for each feedback modality); and subjective evaluation from post-test QoE questionnaires for each of the feedback modalities. In addition, we include analysis by gender.

## 5.1 Objective results

For the objective data, we analysed how the participant reacted to each of the types of feedback i.e. if or how did they change their walking style based on each feedback modality. For each leg, 3 distinct states were defined: varus, correct position, and valgus. We report, for each state, the time the participants remained in misalignment during the experiment, and the number of times the participant needed feedback (feedback cue) during the experiment (2 minutes). We also provide detail on the number of complete alignments (both legs in correct position) and misalignments for each leg.

Table 1 contains performance report of varus and valgus alignment of all participants after experiencing AR and Haptic feedback. It also includes a further categorization by gender. The results show statistically significant differences between the AR and Haptic feedback in terms of the number of varus, valgus, and total misalignments for baseline and test. Participants performed better with AR feedback, with a reduction of 31% for varus, 13% for valgus. All reported results considered 95% and 90% confidence interval. Statistically significant differences in performance is reported for the AR feedback in reducing varus and total misalignments with a two-tailed $p < 0.1$ and $p < 0.05$. For gender analysis, the male improved for varus (45% $p = 0.034$) and valgus (18% $p = 0.073$) while females did not have statistically significant improvement. The ordering of feedback did not influence performance ($p > 0.1$).

**Table 1. Number of varus, valgus and improvement for AR and haptic feedback per gender.**

| Group | Trial | Augmented Reality Feedback | | | Haptic Feedback | | |
|---|---|---|---|---|---|---|---|
| | | Varus | Valgus | Total Misalignments | Varus | Valgus | Total Misalignments |
| Participants | Baseline | 62.772 | 59.363 | 122.136 | 55.273 | 61.545 | 116.818 |
| | Testing | 43.272 | 51.181 | 94.454 | 47.136 | 56.182 | 103.318 |
| | Sig. (2-tailed) | 0.048 ** | 0.444 | 0.046 ** | 0.359 | 0.546 | 0.167 |
| | Improvement | 31% | 13% | 22% | 15% | 9% | 12% |
| Male | Baseline | 76.454 | 74.363 | 150.820 | 71.363 | 67.181 | 138.545 |
| | Testing | 54.818 | 52.000 | 106.818 | 65.272 | 65.000 | 130.272 |
| | Sig. (2-tailed) | 0.034 ** | 0.073 * | 0.041 ** | 0.684 | 0.841 | 0.565 |
| | Improvement | 45% | 18% | 33% | 9% | 3% | 6% |
| Female | Baseline | 49.090 | 44.363 | 93.454 | 39.181 | 55.909 | 95.090 |
| | Testing | 31.727 | 50.363 | 82.090 | 29 | 47.363 | 76.363 |
| | Sig. (2-tailed) | 0.187 | 0.735 | 0.632 | 0.344 | 0.566 | 0.187 |
| | Improvement | 35% | -13% | 13% | 26% | 15% | 20% |

* $p < 0.1$,

** $p < 0.05$

Table 2 contains performance data in terms of how long users were in the varus and valgus positions during the 2 minutes trials. We have confirmed that only AR feedback could reduce varus time with statistically significant difference for baseline and testing. Participants had better performance in time with AR feedback in reducing varus in 11%, valgus 64% and Total misalignments 37%. Males had significant improvement in valgus time (63% p = 0.047). The performance for the Haptic feedback increased the number of misalignments with the male group (-49% p = 0.06). This suggests that the users were somewhat confused by the haptic feedback. Statistically significant difference in performance was only reported for the AR feedback in reducing varus and total misalignments with a two-tailed p<0.05. The ordering of feedback did not influence performance (p > 0.1).

**Table 2. Time of varus and valgus and improvement for AR and haptic feedback per groups.**

| Group | Trial | Augmented Reality Feedback | | | Haptic Feedback | | |
|---|---|---|---|---|---|---|---|
| | | Varus (s) | Valgus (s) | Total Misalignments (s) | Varus (s) | Valgus (s) | Total Misalignments (s) |
| Participants | Baseline | 76.292 | 75.566 | 151.858 | 70.909 | 83.654 | 154.563 |
| | Testing | 67.785 | 26.669 | 94.454 | 85.621 | 75.339 | 160.956 |
| | Sig. (2-tailed) | 0.877 | 0.039 ** | 0.040 ** | 0.142 | 0.348 | 0.635 |
| | Improvement | 11% | 64% | 37% | -21% | 10% | -4% |
| Male | Baseline | 66.504 | 63.934 | 130.438 | 58.515 | 83.204 | 141.720 |
| | Testing | 51.067 | 23.918 | 114.985 | 87.249 | 70.422 | 157.661 |
| | Sig. (2-tailed) | 0.737 | 0.047 ** | 0.439 | 0.060 * | 0.373 | 0.460 |
| | Improvement | 22% | 63% | 12% | -49% | 15% | -11% |
| Female | Baseline | 86.080 | 87.198 | 173.279 | 83.303 | 84.103 | 167.407 |
| | Testing | 84.503 | 72.299 | 156.803 | 83.994 | 80.257 | 164.251 |
| | Sig. (2-tailed) | 0.915 | 0.188 | 0.450 | 0.958 | 0.736 | 0.857 |
| | Improvement | 1% | 17% | 9% | -1% | 5% | 2% |

* $p < 0.1$,

** $p < 0.05$

**Table 3. MOS questionnaire results.**

| QoE Factor | Question | | AR | | Haptic | | |
|---|---|---|---|---|---|---|---|
| | | | MOS | SD | MOS | SD | Sig. (2-tailed) |
| Utility | 1 | *"When I received feedback, I adjusted easily and quickly."* | 4.458 | 1.414 | 3.500 | 1.588 | 0.015 ** |
| | 2 | *"My walking style changed during experiment."* | 4.625 | 1.469 | 5.000 | 1.216 | 0.367 |
| | 3 | *"The system could not be used without the support of an expert."* | 3.083 | 2.205 | 2.708 | 2.331 | 0.362 |
| Usability | 4 | *"The feedback was easy to understand."* | 5.667 | 0.917 | 5.458 | 0.932 | 0.307 |
| | 5 | *"I needed to learn a lot of things before I could use the system."* | 4.625 | 1.377 | 4.875 | 1.191 | 0.366 |
| | 6 | *"The system was difficult to use."* | 5.000 | 1.180 | 4.917 | 1.613 | 0.714 |
| Interaction | 7 | *"The feedback was clear."* | 5.583 | 0.881 | 5.458 | 0.833 | 0.479 |
| | 8 | *"I had to concentrate in order to understand what the system expected me to do."* | 2.542 | 2.167 | 2.042 | 1.944 | 0.261 |
| | 9 | *"The system provided consistent feedback."* | 5.333 | 1.239 | 5.208 | 1.318 | 0.664 |
| Immersion | 10 | *"I was aware of my body whilst moving."* | 5.250 | 1.152 | 5.500 | 1.022 | 0.207 |
| | 11 | *"I was aware of the real world surrounding while walking (e.g. sounds, room temperature, other people, etc.)"* | 1.917 | 2.083 | 1.708 | 1.574 | 0.585 |
| | 12 | *"I was engaged with the system."* | 5.208 | 0.932 | 4.583 | 1.767 | 0.100 |

** $p < 0.05$

## 5.2 Self-reported questionnaire results

Table 3 present results of the MOS self-reported measures via post-test questionnaires. Table IV presents the results considering the gender variable. Since the AR and Haptic groups were randomized repeated measures, a dependent samples t-test was performed on the data with 95% confidence level using the IBM statistical analysis software package SPSS [65].

As per Table 3, out of the 12 questions asked, only Question 1, which was asked if whenever the participant received feedback, he or she adjusted easily and quickly, reported a statistically significant difference between AR and Haptic feedback with a two-tailed p value of 0.015, $p<0.05$. The AR group reported a MOS rating of 4.458 whereas the Haptic feedback 3.5. This result is confirmed that even not knowing performance, participants felt the AR feedback was more effective in reducing misalignments. Considering the discussion in section V.A about how participants responded to the haptic feedback (i.e. increase in misalignments), this results raises an interesting questions about the ease of understanding of haptic feedback for participants. For all other questions, excluding Question 2, the AR feedback had greater MOS than Haptic feedback (although not statistically significant).

Table 4 presents results of the MOS Questionnaire by gender. The female group reported a statistically significant difference between AR and Haptic for Question 1. Male group also reported a statistically significant difference for Question 2 ("My walking style changed during experiment.") and Question 12 ("I was engaged with the system.").

Utility, Usability Interaction, Immersion, and QoEM scores of AR and Haptic feedback by gender are shown in Fig 7. AR feedback showed significant Utility ($p < 0.05$) for female group, which indicated that females found AR feedback more useful than Haptic feedback for this experiment. This QoE factor is related to adjustment to feedback, changes in walking styles and system support.

## 6 Discussion

In this section we discuss the results of the comparison between AR and haptic feedback. Due to the fact that haptic feedback has been reported as a viable feedback modality across many

**Table 4. MOS questionnaire results considering gender.**

| QoE Factor | Question | Male Group | | | | | Female Group | | | | |
|---|---|---|---|---|---|---|---|---|---|---|---|
| | | AR | | Haptic | | | AR | | Haptic | | |
| | | MOS | SD | MOS | SD | Sig. (2-tailed) | MOS | SD | MOS | SD | Sig. (2-tailed) |
| Utility | 1 | 4.417 | 1.564 | 3.667 | 1.723 | 0.169 | 4.500 | 1.314 | 3.333 | 1.497 | 0.049 ** |
| | 2 | 4.000 | 1.758 | 5.250 | 0.621 | 0.044 ** | 5.250 | 0.753 | 4.750 | 1.602 | 0.309 |
| | 3 | 3.083 | 2.353 | 2.667 | 2.424 | 0.318 | 3.083 | 2.151 | 2.750 | 2.340 | 0.653 |
| Usability | 4 | 5.583 | 1.164 | 5.083 | 1.083 | 0.111 | 5.750 | 0.621 | 5.833 | 0.577 | 0.754 |
| | 5 | 4.667 | 1.435 | 4.917 | 1.083 | 0.555 | 4.583 | 1.378 | 4.833 | 1.337 | 0.515 |
| | 6 | 4.833 | 1.403 | 4.583 | 1.781 | 0.491 | 5.167 | 0.937 | 5.250 | 1.422 | 0.777 |
| Interaction | 7 | 5.417 | 1.164 | 5.333 | 0.887 | 0.723 | 5.750 | 0.452 | 5.583 | 0.792 | 0.551 |
| | 8 | 2.333 | 2.229 | 1.583 | 1.729 | 0.212 | 2.750 | 2.179 | 2.500 | 2.110 | 0.718 |
| | 9 | 5.167 | 1.337 | 4.917 | 1.730 | 0.555 | 5.500 | 1.167 | 5.500 | 0.674 | 1.000 |
| Immersion | 10 | 5.167 | 1.466 | 5.250 | 1.356 | 0.754 | 5.333 | 0.778 | 5.750 | 0.452 | 0.175 |
| | 11 | 1.083 | 1.505 | 1.333 | 1.073 | 0.536 | 2.750 | 2.301 | 2.083 | 1.928 | 0.314 |
| | 12 | 5.083 | 1.164 | 3.667 | 2.059 | 0.043 ** | 5.333 | 0.651 | 5.500 | 0.674 | 0.504 |

** p < 0.05

fields such as rehabilitation and gait re-education, our assumption was that haptic feedback would report better results in terms of user performance (and also possibly QoE).

Haptic information is given directly at the joint that the user needs to change whilst AR feedback the participant needed to process visual information and change the leg related to that change. Surprisingly as seen in the results, AR feedback not only reduced the number of misalignments, but from the subjective questionnaire analysis, users reported that AR

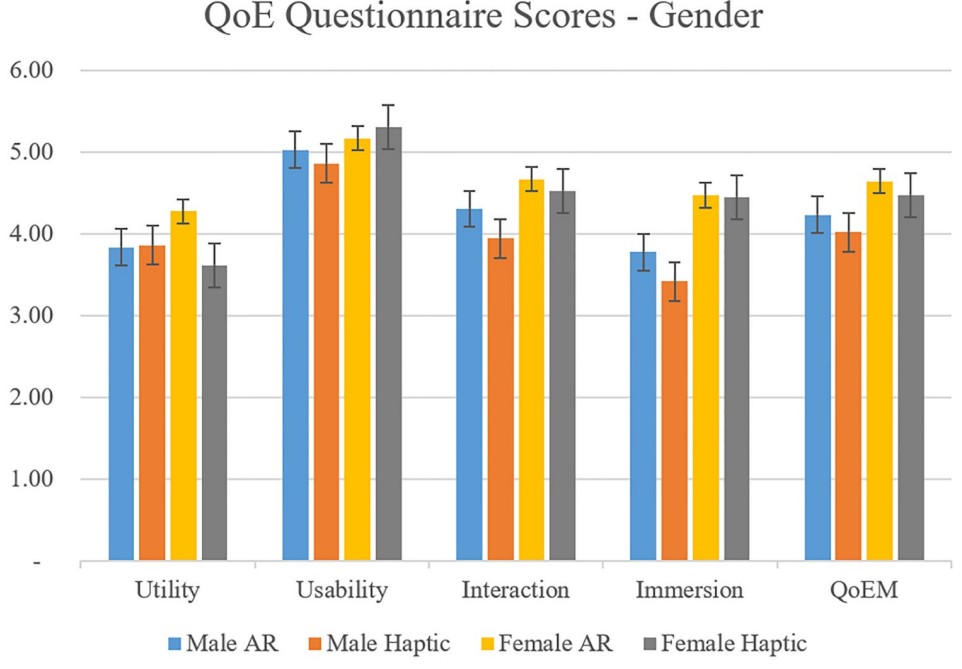

**Fig 7. QoE questionnaire scores for AR and Haptic feedback by gender.**

feedback helped to reduce the number of misalignments better than haptic. Although the results indicate that both feedback modalities reduce the occurrence of varus and valgus misalignments, AR feedback significantly reduced the number of varus misalignment (by 31%) when compared to baseline readings. Whilst the reductions for valgus (for AR) and neither varus nor valgus are significant for haptic, approximate reductions of between 9%-15% are positive.

Looking deeper at the analysis considering gender influence on the results, for the male AR group, the level of reduction for varus was 45% (and 18% for valgus misalignments). Consistent with the male group, although to a lesser extent, AR feedback reduced the number of varus misalignments by 35% for the female group (not significant when compared to baseline). These results demonstrate the utility of employing both feedbacks, but in particular AR feedback. It also raises an interesting question to understand why females' knee did not have a significant change after receiving feedback. Feedback and users' responses to same is an important topic to understand. In our use case, it can have a significant impact on a person's Quality of Life. Reducing misalignments can also reduce the injury incidence more. These results are important for the research community and was also a good indicator for future work, where we will extend the research for understanding physiological measures and what happens in a clinical setup for males and females.

For the QoE analysis, subjective evaluation of questionnaires for feedback utility, usability, interaction, and immersion was performed. Table 3 reported results of the MOS questionnaire for all participants. When participants were asked about adjustment after feedback in Question 1 ("When I received feedback, I adjusted easily and quickly."), they felt that AR was more effective in changing varus and valgus misalignments. This correlates with the objective analysis in Table 1. For the MOS questionnaire considering gender, the male group reported that they believed their walking style changed based on the AR feedback. They also reported higher engagement when using the AR glasses than haptic devices. The female group reported higher utility of AR feedback. These difference between gender groups highlight the importance of considering human factors and employing QoE analysis in these types of novel feedback studies. Considering that many researches were conducted using current feedback tools such as 2D screen and haptic, this study can be a new paradigm in using immersive technologies in gait re-training and promotion of rehabilitation protocols.

## 7 Conclusion

This paper presented a comparison of Haptic and Augmented Reality as feedback modalities in a gait analysis system. It compared, in terms of objective and subjective ratings, how users perceived and responded to Haptic and Augmented Reality feedback. Based on the results, the novel AR approach has significant potential as a method of gait rehabilitation. The objective evaluation tells us that AR significantly reduces the number of knee misalignment. In addition, subjective questionnaire assessment provides interesting results in terms of how users feel their walk changed positively with AR feedback. The agreement of objective and subjective evaluations serves as basis of using AR as part of a rehabilitation protocol. Both gender groups considered reported that AR had greater utility than haptic feedback. The male group showed statistically significant improvement in varus, valgus, total Misalignment, and valgus time. Future work will also assess the validity that AR feedback not only provides higher QoE scores but also promotes less cognitive workload in comparison with haptic as well as instantiation of the QoE model proposed above. Physiologic measures and pupillary response will also be evaluated and their inference to QoE will be analysed.

## Supporting information

**S1 File.**
(RAR)

## Acknowledgments

The authors would like to acknowledge Dr. Paul Archbold and Mr. Eoin Woodlock for the use of the laboratory space for data collection.

## Author Contributions

**Conceptualization:** Thiago Braga Rodrigues, Ciarán Ó Catháin, Noel E. O'Connor, Niall Murray.

**Data curation:** Thiago Braga Rodrigues.

**Formal analysis:** Thiago Braga Rodrigues.

**Funding acquisition:** Thiago Braga Rodrigues, Niall Murray.

**Investigation:** Thiago Braga Rodrigues.

**Methodology:** Thiago Braga Rodrigues, Ciarán Ó Catháin, Niall Murray.

**Project administration:** Thiago Braga Rodrigues.

**Resources:** Thiago Braga Rodrigues.

**Software:** Thiago Braga Rodrigues.

**Supervision:** Thiago Braga Rodrigues, Ciarán Ó Catháin, Noel E. O'Connor, Niall Murray.

**Validation:** Thiago Braga Rodrigues.

**Visualization:** Thiago Braga Rodrigues.

**Writing – original draft:** Thiago Braga Rodrigues.

**Writing – review & editing:** Thiago Braga Rodrigues, Ciarán Ó Catháin, Niall Murray.

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
