## [Decision Letter · Decision Letter 0]

22 Oct 2019

PONE-D-19-27305

A QoE assessment of haptic and augmented reality feedback modalities in a gait analysis system

PLOS ONE

Dear Mr Braga Rodrigues,

Thank you for submitting your manuscript to PLOS ONE. After careful consideration, we feel that it has merit but does not fully meet PLOS ONE’s publication criteria as it currently stands. Therefore, we invite you to submit a revised version of the manuscript that addresses the points raised during the review process.

We would appreciate receiving your revised manuscript by Dec 06 2019 11:59PM. To enhance the reproducibility of your results, we recommend that if applicable you deposit your laboratory protocols in protocols.io, where a protocol can be assigned its own identifier (DOI) such that it can be cited independently in the future. For instructions see: http://journals.plos.org/plosone/s/submission-guidelines#loc-laboratory-protocols

We look forward to receiving your revised manuscript.

Kind regards,

Bijan Najafi

Academic Editor

PLOS ONE

**Journal Requirements:**

**Additional Editor Comments (if provided):**

Thank you for submitting your original study to PLOS ONE. I echo reviewers that the topic of this study is novel and important. However several major weaknesses were pointed out by reviewers that should be addressed before considering your study for publication.

**Comments to the Author**

1. Is the manuscript technically sound, and do the data support the conclusions?

Reviewer #1: Partly

Reviewer #2: No

2. Has the statistical analysis been performed appropriately and rigorously? 

Reviewer #1: No

Reviewer #2: No

3. Have the authors made all data underlying the findings in their manuscript fully available?

Reviewer #1: Yes

Reviewer #2: Yes

4. Is the manuscript presented in an intelligible fashion and written in standard English?

Reviewer #1: Yes

Reviewer #2: Yes

5. Review Comments to the Author

Reviewer #1: In this manuscript, the authors tested effects of haptic and augmented reality feedbacks on varus/valgus motion during walking in healthy young subjects, and evaluated and compared quality of experience between the feedbacks. Please refer to my comments below for improving this manuscript.

Title

Write out QoE. It is very awkward to see an abbreviation in the title: 1) the abbreviation was never introduced in the authors manuscript before, and 2) I am sure there are many potential readers who do not understand what the “QoE” means if they are not written out.

Introduction (and Related Work)

One of the biggest importance of the introduction should be what is the goal of this study, and why is it important to achieve the goal. However, none of these were addressed well in the current introduction. For example, how important giving feedback to patients (e.g., knee oa, arthroplasty) during rehabilitation. Why did authors test haptic and AR feedbacks over other feedbacks? Also, why is it important to evaluate quality of experience? At least, these need to be addressed in the introduction.

For some minor comments, why is “g” always capitalized for “Gait”? Any special reason?

I am not sure if Figure 1 is necessary and appropriate. Before the position of Figure 1, AR and haptic was not even introduced. If the authors want to describe what valgus/varus is for readers who are not familiar with the terms, please revise the figure.

For the Related Work section, I don’t think this section specifically describes the benefits of haptic and AR feedbacks on valgus/varus during walking. The reference [23], testing 9 healthy people does not necessarily “demonstrate” effects of haptic feedback. For AR and QoE, the paragraphs don’t justify why using or assessing these are important.

System and Feedback Architecture

This section provides enough details about data processing. I only have some minor comments.

In Figure 2, please provide photos from some other angles if the authors have some.

In section 3.1., did the authors use matlab and develop their own code for all calculating angles? If so, please state this somewhere. Or did the authors use software that comes with Xsens IMUs (as far as I know, Xsens provides software that calculates angles). If so, I am not sure providing all of these equations are necessary (just simply state “used an embedded algorithm or use Xsens XXX software or something like that”). If the authors want to keep the equations, that is fine but still clarify source of data analysis.

Have the Xsens IMUs been used before for measuring valgus/varus during walking? If so, please add references so that readers know the sensors have been validated for the measurement.

Experimental Protocol

This section is quite poorly written, and more details are necessary for this section. Although some figures show, please described if this is a treadmill walking or something else. Also, how long (or far) was the walking? What was walking speed? Statement regarding obtaining consent form is necessary (even if it is kind of shown in a figure). If the focus is valgus/varus, why were other angles measured? How and when were the feedbacks provided?

I think there should be another section between Experimental Protocol and Results and Analysis in order to explain statistical analysis, QoE analysis. In terns of statistical analysis, for applying for independent t-test, normality check is recommended. If variables did not pass normality test, other statistical methods are recommended.

Results ad Analysis

Please provide some basic demographic information about the subjects. It is difficult to understand tables. For Table 1, are these number varus/valgus? So, the numbers mean the frequency of varus/valgus? Why is it important to report male vs. female? I don’t think gender difference is a focus of this manuscript. Same for the other Tables.

There is no Discussion section?

Reviewer #2: This study targets a very important research topic that is well focused at the intersection of developing a hardware-software system and study of gait analysis-rehabilitation. The authors’ major goal is to propose a system and also a series of objective and subjective metrics in order to help to improve the rehabilitation or corrections for misalignments during walking. As mentioned in the introduction this is an area that requires more research, especially when we consider the availability of state of art AR and haptic technologies. Although the goal and system implementation of this study is well defined and explained in detail, the combination of some key missing information along with lack of explanation of statistical results makes this very interesting manuscript not ready for publication at its current state. Although the paper starts very well written in the first few sections, it leaves the reader with so many critical unanswered questions toward the end. The reviewer summarizes the major concerns/questions of the work along with minor/detailed comments as follows:

Major Comments:

-The most important shortcoming of the study is where the statistical results are presented in tables 1-5.

-Furthermore, there are important conclusions that are clearly inconsistent with the statistical test values reported in table 1-5.

-Overall, the AR feedback condition helps improving subjects’ gait characteristics only in some conditions.

-Haptic feedback surprisingly deteriorates the gait characteristics in some conditions and with a low/debatable statistical significance provides improvement. This is a very important and surprising result that is not explained in the manuscript. Is this a side effect of system itself? The sensors positioning?

-As mentioned in the introduction, the goal of the study is to compare the efficiency of these two modalities. However, considering both objective and subject measures, neither the authors nor the reader can reach a strong conclusion that which one produces a statistically significant difference? Almost all metrics show high p-values.

-How many male/female subjects? It is not clear in what units the values of table 1 are presented. Second, the improvement in AR condition is different for Varus and Valgus in table 1 & 2. Why is this the case? And what should the reader learn from this difference?

-Although many parts of the experiment design, system development, hardware and software details, and coordinate transformations are explained in detail, there is some important missing information that would need some clarifications. For example, were the subjects walking at the same speed? If so, how did this not create a bias in the data collection in terms of the number of gaits for subjects with shorter gait length, given the fact that the testing duration was fixed (2 minutes)?

-A simple temporal diagram seems necessary to clarify the previous question. How long was the baseline data collection? One sample trial of one subject that shows feedback/no-feedback states with theta values would be extremely helpful.

-The model is treated as a black box. Which might be the main cause of the insignificant differences in the results. How does the model transfer a difference in angle or angles into a feedback/no-feedback signal? This is not clearly explained.

-As a user of this system, how would one person know to correct their gait cycle based on a haptic or AR feedback? I tried to imagine this but it seems very arbitrary and unclear how would a subject correct for their gait cycle. Also, the value of 75 seconds out of 120 seconds of testing time shows 62% of the time they were not walking correctly. Is that really the case? This creates an important question about whether the feedbacks are working at all if they were still off 62% of the time. This also raises the question about the model, whether it is tunable to subjects height, wight? Or this is a general model prone to ignoring the between-subject differences?

-QoE questions used in this study seems to be vague or sometimes not suitable for the task. The idea of using QoE, unfortunately, adds more confusion to the already weak statistics presented in Tables 1 & 2. For example, question #10&11 are not very clear for this study since there is not VR display involved and the subjects are of course aware of where their body is located.

-Even the QoE statistics doesn’t help to differentiate between these two modalities of feedback based on the results in table 3,4,5.

-Why do authors think that there is an effect of gender in metrics that are not explicitly related to the perception?

-Due to the mentioned major concerns, the conclusion section would need to go through a major revision along with detail answers for the mentioned questions.

Minor Comments:

(Abstract) Shouldn’t we use the term “subjective” metric? Which is well in contrast with “objective” metrics? As a suggestion, it seems the term “explicit” here is not the best choice.

(Abstract) “Participant’s gait improved” is a vague sentence in the abstract. Even as a general audience the immediate question is: in what way their gait was improved? Are we talking about the timing? Positioning?...

(Line 6 & 27) Since this work is focused at the intersection of the current state of the art technologies (AR and haptics) and the well-established field of gait analysis, I strongly suggest the authors include similar studies where AR projected obstacles are implemented and the performance of the system and the human gait characteristics are reported for the purpose of fair comparison and inclusiveness. Although the references 7-10 are important in the VR/AR literature in general, however, including the contribution of the suggested references (VR/AR+gait analysis) will strengthen the argument and provides an overall picture of the field.

The pickup of visual information about size and location during approach to an obstacle GJ Diaz, MS Parade, SL Barton, BR Fajen - PloS one, 2018

The critical phase for visual control of human walking over complex terrain JS Matthis, SL Barton, BR Fajen - Proceedings of the National Academy of Sciences, 2017

Binaee, Kamran, and Gabriel J. Diaz. "Assessment of an augmented reality apparatus for the study of visually guided walking and obstacle crossing". Behavior research methods. (2018), p.1-9.

(Line 111-112) It is more accurate to mention with minimum or low latency than zero latency, also is there a measurement over the loss of packets on the network? If not, it would be more accurate to say with minimum loss of data rather than “no data is lost”

(Line 135-136) Please clarify that the calibration here is referred to X-Sense IMU system calibration. Which (to my understanding) is assigning a reference coordinate system to the IMU sensors separately.

One of the figures 1,2,3,4 could be used to visualize the theta angles in order to make it easier for the reader to understand in which plane the angles are being calculated and compared with each other.

(Line 149) “is presented” ⇒ “are presented”

(Line 156) Were these haptic feedback modules mounted on the same position for subjects? How did you address the between-subject differences? If this was not a factor that could bias their sensation of haptic (vibration) please briefly mention why do authors think so?

(Line 161-166) The detailed explanation of the circuitry is highly appreciated and valuable for the purpose of reproducing the results or sharing the hardware with other researchers. However, the MOSFET is being used as the switch circuit, therefore, it seems to me the rest of the circuitry (i.e the resistor on the drain-source path) determines the current, hence the intensity of the vibration. Is that correct? If so, the sentence in line 161 seems a little misleading.

(Line 170) in order to be consistent with AR/VR terminology instead of using “Virtual element” please use “Virtual content” (as a suggestion).

(Line 174) Users “Field of View” FOV is also a key factor for an immersive VR/AR experience.

(Line 184) How is the kinematic model generated? Is this a general model or it is modified per subject height, weight, etc.? Please clarify.

(Line 185) Is the threshold value of +-7 in the units of angle? It seems that there’s this hysteresis range between [-7,7] where the implemented algorithm considers the gait to be normal, hence no correction feedback is generated. Right? Please clarify.

If so, would it be useful for the reader to know about the effect of this threshold value? What if, for one subject a threshold of 7 is perfectly fine (in terms of healthy gait characteristics) and for another user threshold value of 4? Although the authors cite a study in this field that addresses the variability, it would be helpful to briefly explain why these threshold values are selected.

One major question/concern regarding the visual presentation of the circles: A very basic and important characteristic of an AR display is to avoid what is referred to as “binocular rivalry” which is presenting two conflicting visual scene to the eyes. This makes the subject experience very negative and causes severe visual discomfort and eye fatigue. Therefore, the diagrams shown in figure 4 are misleading. Is the user seeing 6 total circles? If so this diagram needs to be modified, because it shows that the right and left eye displays are shown conflicting imagery.

Also, in which portion of the field of view these circles are being rendered? What is their size in the visual angle? These are important questions and it will directly affect the users’ perceptual experience as mentioned as one of the major goals of the study.

(Line 203) what is the standard deviation or any measure of variability around mean?

(Figure 5 Caption) “consistent”

(Line 217) Location on which parts of the body?

How long was the training testing session? How fast was the speed of testing/training sessions compared to baseline? How did the baseline data were used?

(Line 237) Response time of feedback with respect to what? Is this a duration? A very important part of the manuscript, yet after reading this section multiple times, I’m not still sure what these metrics are? The authors are encouraged to use a very simple temporal diagram and mark significant events (states) for a sample trial for one subject. This visualization along with a modified explanation seems to be necessary.

What is the “same time interval”? 2 minutes? Please clarify.

Is 2 minutes of data enough to conclude? This is more of a general question rather than concern. Please explain to a more general audience why this provides enough sample for a walking study. Especially, it is not clear how long the feedbacks last on the haptic device or on the display? Do feedbacks immediately disappear as soon as the system reports smaller misalignments relative to the threshold? Or there is a fixed feedback duration time?

A follow-up question/concern: One could argue that the total misalignment might not be the best metric to compare. Because a person with shorter gait, during 2 minutes of walking would provide larger number of gaits compared to someone with a much larger gait length (which is obviously correlated to height). In many similar studies, the adjust/unbias the metric relative to the gait length or gate cycle.

(Line 257) “combination”? ⇒ “Average”

An important question: For both tables 1 and 2, are the reported values averaged over subjects? If so, please also consider reporting the variation from the mean.

Table1: In which physical unit the values are represented? Is this the total angle? Or is this in time? Please mention the number (N) of male and female participants from which the table 1 is generated. Although p-value is an important measure of statistical significance, however, without knowing the effect size the interpretation of the results will be incomplete. There are many ways to calculate this as it is implemented in almost all statistical analysis tools. (https://www.ncbi.nlm.nih.gov/pmc/articles/PMC3444174/)

Comparing Tables 1 & 2: why do we see more improvement in Varus (all subjects) in table 1 but more improvement in Valgus

in table 2? These are the same data from the same subjects, one is represented in the unit of time (table 2) and the other one in the units of degree (I think). This seems to be an inconsistency, right?

In table 2: the negative values show that the feedback mechanism for haptic conditions made the subjects confused or unclear about how they are supposed to correct for their gait. Could this be attributed to the implementation, circuit positioning or there is a perceptual justification for it?

(Line 294) “group”

The choice of questions for this study could be revised. For example, question #10 and #11 is mostly used for a VR experiment in order to check whether the subject would sense a virtual representation of their body parts, such as hand, feet, etc. Therefore, there is no surprise that the two modes of feedback are not perceived differently from the subjects’ point of view.

What does the low MOS value for question #2 tell us? Does it tell anything about the naturalness of the system? Please clarify.

Question #9 Consistent feedback with respect to what? This is a very vague question unless the experimenter has provided additional information to the subject not mentioned in the manuscript. Please clarify.

(Line 298-299) The statistical significance test presented in the table clearly disputes this sentence. The correct conclusion here so far is: the statistical significance test reveals no difference between AR and haptic feedback based on the QoE metric. Except question 1, which could also be ruled out since there is only 1 out of 11. Unfortunately, this is the weakest part of the paper that requires a clear explanation along with the authors’ intuition.

(Line 306) Please avoid using subjective evaluations of the results such as “interesting” as the reader might find these differences very confusing rather than interesting. For example, why do gender matter in #2 which refers to the change in walking? As stated in the table this metric is a utility feature of the system and has very little to do with perceptual judgments (for example being involved with system #12). This seems to be an unanswered question in this study.

(Line 307-313) Please refer to the same comment for lines 298-299. There’s no statistical significance (except a few questions with unclear gender differences).

There are some spaces every few lines that separate the letters of a single word (i.e. lines 230 “discussion”).

6. PLOS authors have the option to publish the peer review history of their article (what does this mean?). If published, this will include your full peer review and any attached files.

Reviewer #1: No

Reviewer #2: Yes: Kamran Binaee

---

## [Author Response · Author response to Decision Letter 0]

17 Dec 2019

Dear Bijan Najafi,

Please find attached my rebuttals to reviewers and updated files to consideration. We have carefully responded to all reviewers comments. We thank you and all reviewers for the effort and help with this manuscript. Their constructive comments and critique have greatly improved the version under review now and we genuinely thank them for this.

Kind regards,

Thiago Braga Rodrigues

---

## [Decision Letter · Decision Letter 1]

12 Feb 2020

PONE-D-19-27305R1

A Quality of Experience assessment of haptic and augmented reality feedback modalities in a gait analysis system.

PLOS ONE

Dear Mr Braga Rodrigues,

Thank you for submitting your manuscript to PLOS ONE. After careful consideration, we feel that it has merit but does not fully meet PLOS ONE’s publication criteria as it currently stands. Therefore, we invite you to submit a revised version of the manuscript that addresses the points raised during the review process.

We would appreciate receiving your revised manuscript by Mar 28 2020 11:59PM. To enhance the reproducibility of your results, we recommend that if applicable you deposit your laboratory protocols in protocols.io, where a protocol can be assigned its own identifier (DOI) such that it can be cited independently in the future. For instructions see: http://journals.plos.org/plosone/s/submission-guidelines#loc-laboratory-protocols

We look forward to receiving your revised manuscript.

Kind regards,

Bijan Najafi

Academic Editor

PLOS ONE

Additional Editor Comments (if provided):

Thanks for addressing the initial concerns. Reviewer #1 has still some concerns that need to be addressed. I however evaluate these concerns to be minor.

Reviewers' comments:

Reviewer's Responses to Questions

**Comments to the Author**

1. If the authors have adequately addressed your comments raised in a previous round of review and you feel that this manuscript is now acceptable for publication, you may indicate that here to bypass the “Comments to the Author” section, enter your conflict of interest statement in the “Confidential to Editor” section, and submit your "Accept" recommendation.

Reviewer #1: All comments have been addressed

Reviewer #2: All comments have been addressed

2. Is the manuscript technically sound, and do the data support the conclusions?

Reviewer #1: Partly

Reviewer #2: Yes

3. Has the statistical analysis been performed appropriately and rigorously? 

Reviewer #1: Yes

Reviewer #2: Yes

4. Have the authors made all data underlying the findings in their manuscript fully available?

Reviewer #1: Yes

Reviewer #2: Yes

5. Is the manuscript presented in an intelligible fashion and written in standard English?

Reviewer #1: Yes

Reviewer #2: Yes

6. Review Comments to the Author

Reviewer #1: Thank you for addressing my comments in this version. I do have some more comments that need to be addressed.

In the introduction, it is still not very clear why AR feedback and Haptic feedback need to be studied. What are the limitations of current feedback tools (2D screens; haptic; audio; expert guidance), and how the current tool that the authors tested those limitations? This needs to be cleared up to justify the need of this study.

In the Methods, so the authors used their own code to calculated the knee angles? Has the code been validated somehow (Even in small sample)?

Discussion needs to be much more deeper. The authors did not really discuss what is expected to be. For example, are the authors' findings expected? Were they in line with previous reports? If so, what does that mean? If not, why were the results different? Also, how and why should we care about these results? AT LEAST, questions above need to be discussed.

Reviewer #2: Thanks to the extensive edits and modifications made by the authors, the core arguments of the paper, the explanations required for an easier understanding of the goals, the analysis and interpretation of the statistical results are significantly improved.

The major questions regarding the manuscript are either addressed in the updated version or explained in the response document.

Going through the revised manuscript there are no major edits seem to be required at its current stage. The efforts put behind revising the manuscript is highly appreciated and it clearly manifests itself in the outcome.

The only minor reminder would be making sure that the quality of figures 5,6,7 are high enough so that after compression it would still look decent.

Good luck with your future research endeavors

7. PLOS authors have the option to publish the peer review history of their article (what does this mean?). If published, this will include your full peer review and any attached files.

Reviewer #1: No

Reviewer #2: Yes: Kamran Binaee

---

## [Author Response · Author response to Decision Letter 1]

27 Feb 2020

Dear Bijan Najafi and reviewers,

Please find attached my rebuttals to reviewers and updated files to consideration. We

have carefully responded to all reviewers comments. We thank you and all reviewers

for the effort and help with this manuscript and its second review. Their constructive comments and critique

have greatly improved the version under review now and we genuinely thank them for this.

---

## [Editor Report · Decision Letter 2]

4 Mar 2020

A Quality of Experience assessment of haptic and augmented reality feedback modalities in a gait analysis system.

PONE-D-19-27305R2

Dear Dr. Braga Rodrigues,

We are pleased to inform you that your manuscript has been judged scientifically suitable for publication and will be formally accepted for publication once it complies with all outstanding technical requirements.

With kind regards,

Bijan Najafi

Academic Editor

PLOS ONE

Additional Editor Comments (optional):

Thanks for your dedication and efforts to address remaining concerns. After reviewing the revision and your response to the remaining concerns, I believe your revision is responsive and the current revision has scientific merit to be published in PLOS ONE. Thus I recommend acceptance of your manuscript. Thanks again for contributing your original study to the PLOS ONE.
---

## [Editor Report · Acceptance letter]

9 Mar 2020

PONE-D-19-27305R2 

A Quality of Experience assessment of haptic and augmented reality feedback modalities in a gait analysis system. 

Dear Dr. Braga Rodrigues:

I am pleased to inform you that your manuscript has been deemed suitable for publication in PLOS ONE. Congratulations! Your manuscript is now with our production department. 

With kind regards,

on behalf of

Dr. Bijan Najafi 

Academic Editor

PLOS ONE